# The Effects of Different Zilpaterol Hydrochloride Feed Supplements and Extended Aging Periods on the Meat Quality of Feedlot Bulls

**DOI:** 10.3390/ani14030361

**Published:** 2024-01-23

**Authors:** Edward C. Webb, Rochelle van Emmenis, Andrew M. Cassens

**Affiliations:** 1Department of Animal Science, Faculty of Natural & Agricultural Sciences, University of Pretoria, Private Bag X20, Hatfield 0002, Pretoria 0028, South Africa; 2Department of Animal Science, Tarleton State University, Box T-0070, Stephenville, TX 76402, USA; cassens@tarleton.edu

**Keywords:** β-adrenergic agonist, confined feeding, beef, tenderness, colour, drip loss

## Abstract

**Simple Summary:**

The aim of this study was to evaluate the effects of β-adrenergic agonist feed additives, namely two commercially available types of zilpaterol hydrochloride (ZH) molecules denoted as ZH-A and ZH-B, on the beef quality of feedlot bulls over extended aging periods. The research stems from concerns that zilpaterol hydrochloride (ZH) may compromise beef quality during exportation by boat as an alternative to the exportation of live animals. Typical South African feedlot bulls were fed a finisher ration supplemented with either ZH-A or ZH-B (both at 105 g ZH/ton) or a negative control (CT) diet during the finishing period. ZH supplementation decreased meat tenderness of both ZH-A and ZH-B treatments compared to the CT when compared at 7, 14, 28, 56 and 120 days of post-mortem aging. ZH supplementation had no effect on meat quality characteristics during post-mortem aging when compared to the CT. The duration of the post-mortem aging period significantly influenced all meat quality characteristics that were investigated, showing improvements in meat tenderness, a decrease in meat colour and an increase in drip and cooking losses from day 56 to 120.

**Abstract:**

This study researched the effects of two commercially available zilpaterol hydrochloride (ZH) β-adrenergic agonists, denoted as ZH-A and ZH-B, on the meat quality characteristics of typical South African feedlot bulls (taurine × indicus composites), over extended aging periods of up to 120 days. The effects of ZH were studied to address concerns about the possible adverse effects of ZH on beef quality following extended aging, which typically occurs during the exportation of beef by boat. The completely randomized control study consisted of 3 homogenous experimental groups, with 3 replicates per treatment and 50 bulls per replicate = 450 animals. Treatments were a negative control (CT) with no ZH supplementation added to the basal diet or a basal diet supplemented with either zilpaterol hydrochloride A (ZH-A) or zilpaterol hydrochloride B (ZH-B), both at 105 g ZH/ton, fed from the first day of the finishing period (D_0_) for 30 days. Subsamples were collected from 38 random carcasses from each treatment for proximate analysis and meat quality analysis using Longissimus dorsi samples. ZH supplementation decreased meat tenderness (Warner–Bratzler shear force values (WBSF)) of bulls fed ZH-A or ZH-B, compared to those fed the CT diet (*p* < 0.05; η^2^ = 0.24). The WBSF values of both ZH treatments were about 0.5–0.8 kg higher during the aging periods compared to the CT, but ZH did not affect post-mortem meat aging or meat quality characteristics differently compared to the CT. Post-mortem aging per se influenced all meat quality characteristics investigated (*p* < 0.001; η^2^ > 0.30), showing improvements in WBSF, a decrease in meat colour and an increase in drip and cooking losses. L*-values increased from 3 to 56 days of aging and then decreased to day 120 (*p* < 0.001; η^2^ = 032). Chroma values decreased from day 3 to day 120 (*p* < 0.001; η^2^ = 0.50). Hue° decreased from day 3 to day 7 and stabilized until day 120 (*p* < 0.001; η^2^ = 0.40). Moisture and cooking loss (CL) increased to 56 days and then decreased to 120 days.

## 1. Introduction

There are growing concerns about the projected world population growth [1] and the sustainability of agriculture in terms of its environmental impact, the use of natural resources and the ethics of animal source food production [2,3]. The demand for food is projected to increase markedly [4], with subsequent effects on employment rates, job creation, cost of living and the socioeconomic welfare of citizens [2,5,6]. There is growing pressure on low- and middle-income countries (LMICs) to be more sustainable, but improvements in crop production are generally limited by climatic conditions, access to natural resources or geography. In contrast, livestock farming is one of a few sustainable and economically viable options to produce nutrient-dense foods, compared to vegan/vegetarian alternatives [2,5,6].

A growing number of consumers have a negative perception of livestock production, but in most countries, animal agriculture is making concerted efforts to minimize the environmental impact and improve resource use to ensure sustainability and to gain consumer support for animal source foods [3,6]. With better production management and the use of growth-enhancing technologies, fewer animals are required to produce the same amount of kg product, which also has welfare benefits [3,4]. The use of feeding strategies and exogenous growth-enhancing molecules improves production efficiency and can reduce the environmental impact of animal agriculture [4,7,8,9,10]. Currently, countries such as the European Union (EU) and Russia have legislation in place (e.g., 96/22/EC) that ban the importation of products that were produced using growth-enhancing molecules such as β-adrenergic agonists (βAs) and steroidal growth implants [11,12]. The South African beef industry currently uses such molecules to supply its own demand for beef, and exports to neighbouring countries and China are increasing [5].

Food distribution via live animal exportation is also attracting attention due to animal welfare concerns. The National Council of the Society for the Prevention of Cruelty to Animals (NSPCA) and consumers in South Africa are concerned about the exportation of live animals, especially following recent incidents like the exportation of about 80,000 live sheep from the Eastern Cape, South Africa to Kuwait, Western Asia via boat in 2019. This led to high animal, product and financial losses and raised concerns about the ethics of live animal exports. The export of carcasses and selected meat cuts is a more practical and ethical approach. Unfortunately, concerns were raised about the effects of exogenous growth-enhancing molecules such as zilpaterol hydrochloride on beef quality after extended aging periods, such as during transportation by boat.

The objective of the present study was to investigate the effects of two commercially used zilpaterol hydrochloride β-adrenergic agonists, denoted as ZH-A and ZH-B (both registered for use in South Africa), compared to a negative control treatment on beef quality and shelf life (from 24 h to 120 days post-mortem, including a short retail display). Meat tenderness, drip loss, cooking loss, colour stability and shelf life are important quality attributes of beef [13,14], which may discourage the exportation of beef if compromised.

## 2. Materials and Methods

### 2.1. Ethical Approval and Treatments

Ethical approval for this research was obtained from the Research Ethics Committee of the University of Pretoria with the ethics approval number NAS390/2019. The current research formed part of a larger research study on the effects of beta-adrenergic agonist feed additives on the growth performance, carcass and meat quality of typical South African feedlot cattle. The present experiment focused specifically on the effects of the feed supplementation with two different types of commercially available zilpaterol hydrochloride (ZH) molecules (denoted as ZH-A and ZGH-B) during the finishing period of typical feedlot bulls on the meat quality characteristics and shelf life of meat up to 120 days post-mortem. The active ingredient of both beta-adrenergic agonists is zilpaterol hydrochloride, and both were supplemented in the feed during the last 30 days of the finishing period.

### 2.2. Experimental Design and Animals

The experimental design was a completely randomized control study, consisting of 3 experimental groups, 3 replicates per treatment group and 50 bulls per replicate (3 × 3 × 50 factorial design). The research study was conducted at a large commercial beef feedlot located in Gauteng, South Africa. A total of 450 homogenous intact bulls were randomly selected from a total of 2000 typical intact feedlot bulls made available for this study. Composite-type bulls of medium size and maturity (e.g., Bonsmara crosses) were used. The cattle were representative of typical feedlot cattle in South Africa, which were all identified with durable Allflex^®^ (MSD Animal Health, Somerset West, South Africa) ear tags. The average initial mass of the bulls was ca. 400 kg when the experiment commenced (D_0_).

All bulls were processed and managed similarly to limit variation. This included preventative treatments and vaccinations for internal and external parasites, injecting vitamin supplements and immunization against Clostridial and viral diseases, including respiratory diseases such as infectious bovine rhinotracheitis (IBR). During processing, all cattle received an initial hormonal growth ear implant, i.e., Ralgro^®^ (MSD Animal Health, Somerset West, South Africa) and were then re-implanted with Revalor H^®^ (MSD Animal Health, Somerset West, South Africa) after 45 days. Revalor H^®^ was then effective for another 60–70 days. All bulls were housed in large soil-surfaced floor pens (ca. 50 m^2^ standing space/bull) and fed the same starter and grower diet after an adaption period of 14 days. Fourteen days prior to the normal finishing period (D-14), the cattle were weighed using a heavy-duty Richter electronic weight beam scale and randomly allocated to different treatments to commence (D_0_) the finishing period.

### 2.3. Allocation of Experimental Treatments

The treatments were administered from D_0_ for 30 days during the normal finishing period, followed by a 3-day compulsory ZH-withdrawal period before slaughter (D_33_), to comply with regulations of the use of β-adrenergic agonists in South Africa. The treatments were administered to the experimental groups during the feedlot finishing period, i.e., from D_0_ to D_30_. The treatments were a negative control (CT) which received no zilpaterol hydrochloride (ZH) supplementation, the second was ZH-A feed supplementation (ZH-A at 105 g ZH/ton) and the third was ZH-B feed supplementation (ZH-B at 105 g ZH/ton).

### 2.4. Feeding and β-Agonist Feed Supplementation

Following standard feedlot operating procedures, the animals were fed in bulk feed troughs that ensured adequate feeding space per animal (ca. 30 cm feeding space/bull). The animals received a standard, balanced, dry feedlot concentrate ration ad libitum that provided 10.5 MJ ME/kg DM energy during the finishing phase of feeding. The daily allotment of feed for each pen was estimated and recorded via bunk-reading the feed consumed from that which was provided. The expected or predicted feed intake was calculated using an estimate of roughly 3% of the live weight (kg) being consumed per day for live performance analyses. The animals had ad libitum access to good-quality water. Supplements were mixed into the 75% DM content feed rations at a standard recommended concentration of 105 g of ZH/ton. Every batch of finishing ration was sampled and bio-assayed to confirm the β-agonist concentrations.

### 2.5. Slaughter, Meat Sample Collection and Storage

After a feedlot finishing period of 30 days and a 3-day compulsory withdrawal period, the final mass (D_33_ mass) of the bulls was recorded, and all the bulls were loaded and transported to the commercial abattoir in Gauteng, South Africa. After 24 h of lairage, the animals were slaughtered in the conventional way (e.g., slaughter after stunning). This involved the mechanical stunning of bulls using a pneumatic stunner (captive bolt), followed by suspension from their hocks and exsanguination. The carcasses were electrically stimulated (110 V, 17 Hz, 5 m/s; Jarvis Product Corporation, Alberton, South Africa, Pty Ltd.) during hoisting, followed by evisceration and skinning [12,15].

Of the 450 carcasses, 38 were randomly selected per experimental group for meat quality analyses (38 × 3 treatments = 114 carcasses). The carcasses were placed in a chiller (2–4 °C) at approximately 45 min post-mortem. Muscle samples were collected from the carcasses after chilling for 24 h. Samples were excided from the Longissimus dorsi muscle (LD; muscle generally used for meat science studies) between the 9th and 12th ribs on the left side of each carcass [12,15]. Muscle samples were labelled, vacuum packed and stored at −1 to 0 °C for different aging periods (7, 14, 28, 56 and 120 days) until meat quality analyses were performed.

### 2.6. Meat Quality Analyses

The following meat quality evaluations were performed after LD samples were thawed at 4 °C for 36 h after aging periods of 7, 14, 28, 56 and 120 days. Meat samples were placed at 25 °C (room temperature) for 20 min to bloom (i.e., turning bright red due to oxygen exposure) for colour measurements using a colorimeter—Hunter Lab ColorFlex EZ (Model 45/0 LAV, Hunter Laboratory Associates, Inc., Reston, VA, USA) [16]. According to the manufacturer’s guidelines, the ColorFlex EZ was calibrated immediately before readings against black and white tiles. L* (lightness), a* (redness) and b* (yellowness) values were determined utilizing a standard observation made at 10° using illuminant D65 (with an accuracy of ∆E* < 0.15 CIE L*a*b* (Avg) on BCRA II Tile Set). Three random readings were taken to obtain a mean value recorded. Chroma (red colour intensity) and hue angle (meat discoloration) values were calculated [16] as follows:*Chroma* (*C**) = (*a*^*2^ + *b*^*2^)^1/2^
*Hue angle* (*h*°) = *tan* − 1 ((*b**)/(*a**)).

Radiant values were converted to degrees using multiplication by 57.2958. Moisture (drip) loss was calculated as the percentage mass of fluid that was lost from meat samples during storage, and cooking loss was determined as previously described [17]. Meat samples were weighed and then placed in a plastic bag to be cooked in a water bath at 80 °C until the internal temperature of meat samples was 70 °C. The cooked samples were then chilled overnight at 4 °C, blotted dry and weighed to calculate the weight loss after cooking. Cooking loss was calculated as a percentage of the initial mass by (mass loss after cooking)/(initial sample mass) × 100%. Meat tenderness was tested on the same samples used for CL measurements via a shear force (WBSF) test performed with an Instron apparatus equipped with a Warner–Bratzler shear blade (G-R Elec. Mfg. Co., Manhattan, KS, USA). The WBSF measurement was used as a physical indicator of meat tenderness instead of a perceived sensory panel score. Core meat samples (2.54 × 10 cm) were excised from the cooked meat samples to measure the mean peak force value (kg) to cut the meat perpendicularly to the muscle fibres. The mean WBSF was derived from five peak force recordings.

### 2.7. Statistical Analyses of Data

Data were recorded in Excel and then imported into to IMB SPPS Statistics, Version 28, 2022 (New York, NY, USA). ZH supplementation and differences between the two ZH molecules were investigated using General Linear Model (GLM) analyses, and significant differences were recorded at *p* < 0.05. Because the number of bulls differed marginally between the replicates and experimental groups (a bull jumped the fence on D_0_ and remained there), Bonferroni’s multiple range test was used to analyze the differences between least squares means in an unbalanced experimental design for both pooled and pairwise comparisons between treatments. The analysis of the effects of extended post-mortem aging on beef tenderness and colour parameters of ZH and control groups were carried out via Multinomial Logistic regression analyses, with the aging period and ZH treatment as factors, and each meat quality attribute as reference categories.

## 3. Results

The growth and carcass results of the feedlot bulls supplemented with zilpaterol hydrochloride (ZH) compared to the negative control (CT) treatment are summarized in Table 1. The average mass of bulls at the start of the finishing phase (D_0_) was similar due to randomization, but slaughter mass was higher (*p* < 0.05; Table 1) for the bulls supplemented with ZH compared to the bulls not supplemented. Similarly, ZH supplementation increased the carcass mass (*p* < 0.01) and dressing percentage compared to CT. The percentage of carcass mass loss at 24 h post-mortem was higher (*p* < 0.01) for the carcasses from the ZH treatment compared to those from CT, probably due to the lower subcutaneous fat thickness (*p* < 0.02) of the ZH carcasses.

The effects of feed supplementation with either ZH-A or ZH-B, compared to CT, on meat quality attributes of the composite-type feedlot bulls of medium maturity during the finisher period are presented in Table 2.

There was no difference between the two types of zilpaterol hydrochloride products, namely ZH-A, compared to ZH-B on the meat quality attributes of LD meat samples. However, LD samples from the bulls supplemented with either ZH-A or ZH-B (collectively referred to as ZH) had higher shear force values (*p* < 0.001; Table 2) and were therefore less tender compared to those from the control treatments. The colour attributes of the meat samples did not differ between the CT and ZH groups, but cooking loss tended to be lower (*p* = 0.057) in samples from the ZH-A group compared to the control group. There were no significant differences in meat quality characteristics between ZH-A vs. ZH-B, as determined at all of the post-mortem aging periods of 7, 14, 28, 56 or 120 days in this study, indicating similar meat colour, moisture, cooking losses and WBSF (Table 2). For this reason, the effects of extended post-mortem aging periods of the CT compared to pooled ZH treatments on meat quality parameters are presented in Table 3.

ZH treatments mainly affected meat tenderness, while aging periods affected beef colour, moisture and cooking losses (Table 3). Comparisons between WBSF shear force between ZH and CT were small at 7 days of aging but differed at extended aging for 14, 28, 56 and 120 days (*p* < 0.01). Although shear force differed at 120 days of aging (*p* < 0.05), the tenderness from all treatments was best at about 56 days of aging. WBSF values decreased, and therefore the tenderness of the LD samples improved with increasing aging periods from 7 to 120 days post-mortem, but extended aging did not completely negate the meat toughing effects of ZH supplementation in this study. Fortunately, meat samples from all treatments were classified as tender in terms of the tenderness threshold of about 7 kg or 70 N, which occurred after 7 days of aging [18].

No significant differences in any of the meat colour parameters tested, nor for moisture loss of samples from CT and ZH groups, were observed at any of the aging periods tested. Post-mortem aging period influenced (*p* < 0.01) meat colour parameters, moisture and cooking losses, with marginal differences between CT and ZH treatment groups but with similar trends over time.

## 4. Discussion

Zilpaterol hydrochloride (ZH) supplementations (ZH-A or ZH-B) in the feed during the finisher period increased slaughter mass, cold carcass mass and dressing percentage of feedlot bulls compared to the negative control groups (CT), which agrees with previous studies [7,8,9,15,18,19,20,21,22,23,24,25]. Subcutaneous carcass fat thickness was reduced in ZH-supplemented bulls, which resulted in higher carcass weight losses post-mortem compared to the CT. Similar findings were reported in previous studies, indicating that zilpaterol hydrochloride supplementation reduces carcass fat content in intensively fed cattle [7,8,9,15,18,19,20,21,22,23,24,25].

No differences were observed between the two ZH supplementation groups (i.e., ZH-A vs. ZH-B; *p* > 0.40) for any of the meat quality characteristics. A previous study also found no significant meat quality differences; although some trends in chroma and hue angle were observed [8], those results were in zebu cattle, while the present study was in taurine x zebu composites. Overall, ZH supplementation increased WBSF values (hence decreased tenderness) as tested at all aging days, compared to samples from the CT treatment (*p* < 0.05; η^2^ = 0.24).

In one previous study [18], ZH supplementation increased WBSF values by 22% in feedlot steers compared to negative controls. In studies where WBSF was measured over 21 days, ZH supplementation increased WBSF values compared to untreated controls [19,20]. Several previous studies [9,12,19,20,23,24] indicated that post-mortem aging significantly improved tenderness by decreasing WBSF. It was also reported [25] that the increase in WBSF in ZH-supplemented cattle is due mostly to muscle hypertrophy and that longer aging periods may be required to ensure acceptable meat tenderness.

For this study, extending the aging period of LD samples from feedlot bulls to 120 days post-mortem had a large, significant effect (*p* < 0.01) on all meat quality characteristics measured. Although the aging of the meat samples significantly decreased WBSF values for both CT and ZH, the aging effect size for WBSF of samples from ZH was smaller compared to that from the CT treatments (η^2^ = 0.18 vs. 33). The regression equations for WBSF values over extended post-mortem aging periods for samples from ZH and CT, are presented in Figure 1. Although these regression equations indicate a decrease in WBSF over extended post-mortem aging periods, WBSF decreased faster in CT samples and differed (*p* < 0.05) compared to ZH samples from 14 days post-mortem onwards.

No significant differences were observed between ZH-A and ZH-B for L*, a*, b*, chroma or hue° values of beef samples during specific aging periods. These observations generally agree with previous research [7]. However, L*-values (brightness) were similar, but a* (redness) and b* (yellowness) values were lower for all treatments compared to the previous study on the same molecules [7], which may reflect breed type and dietary effects like the type of roughage used [26]. Although no differences were observed in meat colour attributes between ZH and CT, apart from chroma at 56 days (*p* < 0.01) and a trend for hue° at 14 days (*p* < 0.1) of aging, the multinomial logistic regression analyses of WBSF and colour attributes over post-mortem aging days, revealed differences (*p* < 0.01) between the CT and ZH treatments.

The multinomial logistic regressions for WBSF and colour attributes over extended aging periods, as affected by CT vs. ZH treatments, are presented in Figure 2a–f. These regression curves reflect lower values for WBSF and meat brightness for CT over extended aging periods, while meat redness (a*), yellowness (b*), hue°, and colour saturation (chroma) reflect lower curves for samples from ZH. So, although the multinomial logistic regressions confirm the beneficial effects of aging on WBSF, lower a*, b* and red colour intensity in ZH-supplemented bulls was more apparent, while the risk of meat discoloration (higher hue°) was higher for CT bulls with extended aging.

No differences were observed in moisture losses between CT and ZH treatments, while a trend for lower cooking losses (*p* = 0.057) was observed for samples from ZH. Both CT and ZH treatments had the highest moisture losses at 56 and 120 days of aging (*p* < 0.01), while cooking losses were highest at 28 and 56 days of aging. Moisture losses over extended aging periods were due to the combined effects of muscle fibre structure damage (via protein degradation), which releases intracellular fluids and chromatic pigments (mainly myoglobin) in the moisture losses [27].

Previous studies [18,19] reported no ZH-supplementation effects on cooking losses, but those studies only include sampling up to 21 days post-mortem. Structural muscle damage over extended aging periods may influence moisture and cooking losses, water-holding capacity and WBSF values [27]. In the present study, moisture loss explained a significant portion of the variation in WBSF (R^2^ = 0.15, *p* < 0.01) for samples of both CT and ZH treatments.

## 5. Conclusions

It was demonstrated that bulls fed the β-adrenergic agonists ZH-A or ZH-B containing zilpaterol hydrochloride (ZH) for 30 days significantly decreased meat tenderness due to increased WBSF values. Dietary supplementation with ZH in feedlot diets during the finishing period did not affect the meat quality differently during extended aging compared to the negative controls. No significant differences in meat quality were observed between ZH-A and ZH-B. The extended aging of meat samples significantly influenced all meat quality characteristics of feedlot bulls in all experimental groups. Although meat tenderness was initially negatively affected by ZH supplementation, it improved significantly over extended post-mortem aging. ZH supplementation per se does not affect post-mortem meat quality more than untreated bulls, but extended aging may compromise beef quality if it exceeds 56 days.

## Figures and Tables

**Figure 1 animals-14-00361-f001:**
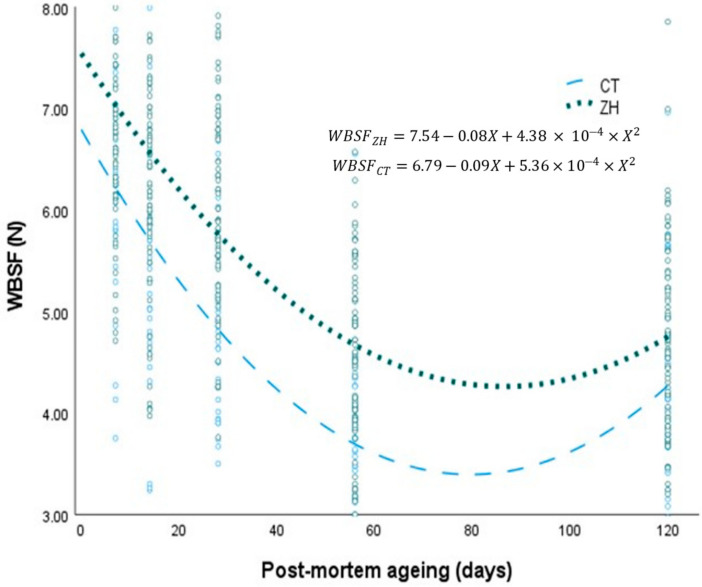
Effects of post-mortem aging on beef tenderness (WBSF) of feedlot bulls supplemented with zilpaterol hydrochloride (ZH) compared to negative controls (CT), (°° Data points).

**Figure 2 animals-14-00361-f002:**
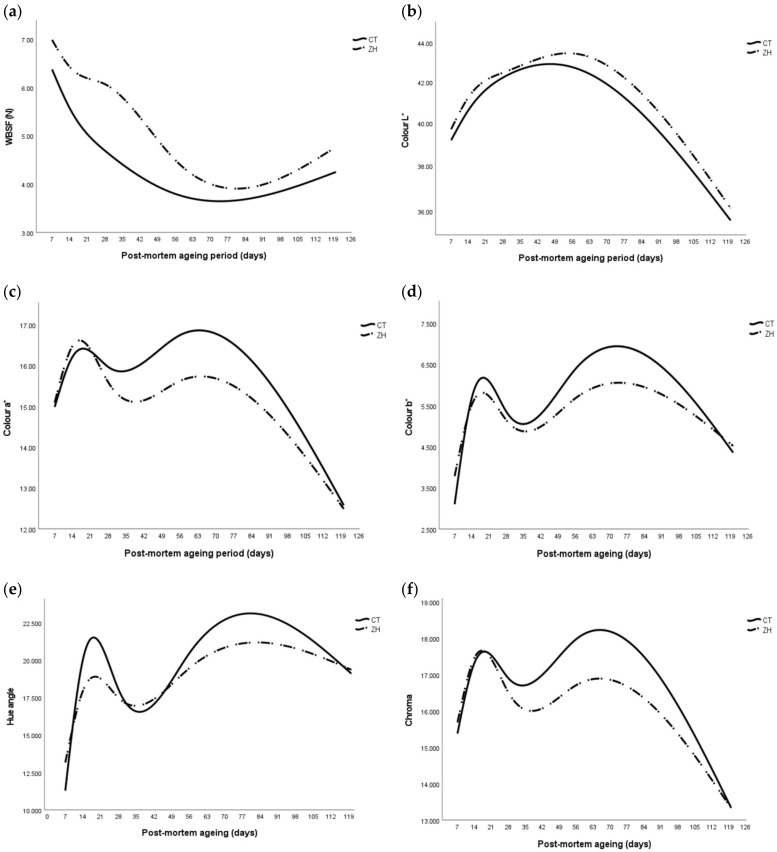
Multinomial logistic regression analyses for beef (**a**) WBSF and colour attributes (**b**) L*, (**c**) a*, (**d**) b*, (**e**) hue° and (**f**) chroma over extended aging periods for samples of bulls supplemented with zilpaterol hydrochloride compared to controls.

**Table 1 animals-14-00361-t001:** Effects of feed supplementation with zilpaterol hydrochloride (ZH) during the finishing period compared to negative control (CT) treatment on the growth and carcass characteristics of composite-type feedlot bulls.

Growth Characteristic	Control (CT) Mean (Std Error)	Zilpaterol (ZH) Mean (Std Error)	*p* = F
D_0_ mass (kg)	397.97 (1.843)	397.58 (1.303)	0.86
Slaughter mass (kg)	420.83 ^a^ (1.267)	425.95 ^b^ (0.896)	0.05
Cold carcass mass (kg)	249.40 ^a^ (0.903)	256.52 ^b^ (0.638)	0.01
Carcass mass loss %	2.43 ^a^ (0.091)	2.74 ^b^ (0.065)	0.01
Dressing %	59.23 ^a^ (0.180)	60.23 ^b^ (0.127)	0.01
SC fat (mm)	5.77 ^a^ (0.158)	5.31 ^b^ (0.112)	0.02

^a,b^ Means with different superscript letters differ (*p* < 0.01); D_0_ mass—mass at start of finishing period; SC fat—Subcutaneous fat thickness over 13th thoracic vertebrae, 5 cm from the medial line. ZH—Zilpaterol hydrochloride treatment (pooled).

**Table 2 animals-14-00361-t002:** Effects of the supplementation of feedlot bulls during the finisher phase with β-adrenergic agonists ZH-A, ZH-B or negative control (CT) on meat quality attributes.

Meat Quality Variable	Control (CT) Mean (Std Error) (*n* = 162)	ZH-A Mean (Std Error) (*n* = 172)	ZH-B Mean (Std Error) (*n* = 166)	*p* = F
WBSF_N	4.79 (0.110) ^a^	5.65 (0.105) ^b^	5.61 (0.121) ^b^	0.001
Colour L*	40.15 (0.296)	40.93 (0.283)	40.72 (0.289)	0.147
Colour a*	15.30 (0.170)	15.01 (0.163)	15.12 (0.166)	0.471
Colour b*	4.99 (0.163)	4.85 (0.156)	5.03 (0.159)	0.687
Hue°	18.60 (0.524)	18.30 (0.467)	17.62 (0.477)	0.567
Chroma	16.18 (0.196)	15.87 (0.188)	16.02 (0.191)	0.517
H_2_O-loss (%)	7.59 (0.203)	7.74 (0.194)	7.89 (0.198)	0.564
Cooking loss (%)	23.57 (0.275) ^x^	22.67 (0.263) ^y^	22.97 (0.268) ^xy^	0.057

^a,b^ Means with different superscript letters differ (*p* < 0.01); ^x,y^ Means with different superscript letters tend to differ (*p* < 0.1).

**Table 3 animals-14-00361-t003:** Effects of extended meat aging periods (7-, 14-, 28-, 56- and 120 days) and zilpaterol hydrochloride treatment (ZH~ZH-A + ZH-B pooled), compared to negative controls on the meat quality attributes of Longissimus dorsi samples from feedlot bulls.

Meat Quality Variable		Control Treatment Mean (Std. Error)	ZH Treatment Mean (Std. Error)
WBSF ^#^ (kg)	7	6.62 (0.278) ^a^	7.12 (0.186) ^a^
	14	5.57 (0.188) ^b,A^	6.43 (0.132) ^b,B^
	28	4.70 (0.188) ^c,A^	6.07 (0.130) ^b,B^
	56	3.80 (0.188) ^d,A^	4.50 (0.132) ^c,B^
	120	4.25 (0.191) ^cd,A^	4.77 (0.131) ^c,B^
Colour L*	7	39.26 (0.891) ^a^	40.15 (0.586) ^a^
	14	40.72 (0.586) ^b^	41.43 (0.414) ^ab^
	28	42.29 (0.594) ^b^	42.67 (0.409) ^bc^
	56	42.80 (0.586) ^b^	43.62 (0.412) ^c^
	120	35.68 (0.594) ^c^	36.27 (0.409) ^d^
Colour a*	7	14.99 (0.512) ^a^	15.17 (0.337) ^ab^
	14	16.22 (0.337) ^a^	16.47 (0.238) ^b^
	28	15.97 (0.342) ^ab^	15.57 (0.235) ^ab^
	56	16.75 (0.337) ^b^	15.63 (0.237) ^ab^
	120	12.58 (0.342) ^c^	12.49 (0.235) ^c^
Colour b*	7	3.13 (0.489) ^a^	3.86 (0.322) ^a^
	14	5.70 (0.322) ^b^	5.48 (0.228) ^b^
	28	5.36 (0.326) ^bc^	5.13 (0.225) ^bc^
	56	6.39 (0.322) ^b^	5.68 (0.226) ^b^
	120	4.36 (0.326) ^ac^	4.52 (0.225) ^ac^
Chroma	7	15.38 (0.590) ^a^	15.77 (0.388) ^a^
	14	17.32 (0.388) ^ab^	17.44 (0.274) ^b^
	28	16.91 (0.393) ^ab^	16.49 (0.271) ^ab^
	56	17.97 (0.388) ^b,A^	16.71 (0.273) ^ab,B^
	120	13.33 (0.393) ^c^	13.32 (0.271) ^c^
Hue°	7	11.29 (1.462) ^a^	13.17 (0.978) ^a^
	14	20.10 (0.991) ^b,C^	17.87 (0.696) ^b,D^
	28	18.07 (0.991) ^b^	17.50 (0.687) ^b^
	56	20.49 (0.991) ^b^	19.49 (0.696) ^b^
	120	19.12 (1.004) ^b^	19.38 (0.691) ^b^
H_2_O_loss (%)	7	4.88 (0.609) ^a^	4.81 (0.400) ^a^
	14	6.13 (0.400) ^ab^	6.73 (0.283) ^b^
	28	7.02 (0.406) ^b^	7.46 (0.279) ^b^
	56	10.69 (0.400) ^c^	10.86 (0.281) ^c^
	120	9.22 (0.406) ^c^	9.22 (0.279) ^d^
Cooking loss (%)	7	4.88 (0.826) ^a^	4.81 (0.543) ^a^
	14	28.10 (0.543) ^b^	27.75 (0.384) ^b^
	28	30.57 (0.551) ^b^	29.85 (0.379) ^c^
	56	30.62 (0.543) ^b^	29.80 (0.381) ^c^
	120	22.79 (0.551) ^c^	21.89 (0.379) ^d^

^a,b,c,d^ Means with different superscript letters in the same column (between aging periods) differ (*p* < 0.05); ^A,B^ Means in rows (between treatments) differed (*p* < 0.01); ^C,D^ Means in rows (between treatments) tended to differ (*p* < 0.1); ^#^ WBSF—Warner–Bratzler shear force (kg).

## Data Availability

Data are contained within the article.

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
