# Peer review of "The Effects of Different Zilpaterol Hydrochloride Feed Supplements and Extended Aging Periods on the Meat Quality of Feedlot Bulls"

_animals, 2024, doi:10.3390/ani14030361_

Round 1
Reviewer 1 Report
Comments and Suggestions for Authors
In this manuscript, the authors investigate the effects of zilpaterol hydrochloride (ZH-A and ZH-B) and post-mortem ageing on meat quality characteristics of feedlot bulls. The authors found that dietary supplementation with ZH-A and ZH-B significantly decreases meat tenderness compared to control groups. Post-mortem ageing significantly improves meat tenderness, and ZH-supplementation has no effect on meat quality characteristics during post-mortem ageing compared to control groups.
This manuscript is very straightforward and concise. The authors did a good job using multiple matrices to evaluate meat quality, including meat color, WBSF, Drip loss, and cooking loss. The authors also conducted a very detailed statistical analysis of the data. I also appreciate the discussion section where the authors compare the current study with previous results, providing rationale and hypotheses for each result.
However, I think the paper could benefit from a few minor improvements.
-
The connection between zilpaterol hydrochloride supplementation and extended post-mortem ageing is not clearly stated in the manuscript. The authors state that they want to investigate the effect of growth-enhancing modules on quality and shelf life. The connection between shelf life and post-mortem aging needs to be further illustrated.
-
I wish there were more discussions on the significance of the results of the manuscript.
Author Response
- The manuscript has been significantly edited and several part rewritten completely.
- The connection between zilpaterol hydrochloride and extended ageing periods has been better explained and put in context to the purpose of the manuscript.
- The connection between post-mortem ageing and shelf life has been explained better and linked to the respective meat quality attributes assessed in the present study.
Reviewer 2 Report
Comments and Suggestions for Authors
Dear authors,
The manuscript animals-2776606, entitled "Effects of different zilpaterol hydrochloride feed supplements and extended ageing periods on the meat quality of feedlot bulls" deals with an interesting and relevant for the "Carcass Traits and Meat Quality in Cattle" Special Issue.
To assess its suitability for publication in Animals, I performed a thorough review of each section of the manuscript. General comments and recommendations for each section are provided below:
Introduction: The "Introduction" presented the importance of conducting this study in an organized manner. However, it was presented in a very simplistic way, with little detail and few references on the subject (only 11, of which 3 are self-citations). Therefore, it is necessary to improve it by providing more detail on the subject and highlighting the main advances in research around study in a more clear and concise manner. At the end of this section, the research hypothesis, and the objective of the study (in accordance with the scientific method) should also be clear. I strongly recommend that this section be completely rewritten.
Materials and Methods: The "Materials and Methods" section is well-structured, but it needs more detail in terms of the methods used. It is important to include more information in this section about the methodologies used, as well as the equipment used, so that the reader can understand in depth what was done. Therefore, I recommend that it be improved, bringing more detail.
Results: The results are presented in a partially adequate way. The tables need to be improved to make them more self-explanatory and easier to understand. Additionally, this section should not discuss these results, leaving this for the next section (Discussion and Conclusions). I recommend that it be completely reorganized to make it appropriate to what was proposed and easier for the reader to understand.
Discussion: The "Discussion" section was presented in a partially adequate way. However, it would be beneficial to further explore the discussions based on the reading and/or to insert more inferences about the results obtained. I recommend that this be done.
Conclusions: The conclusions reached were supported by the experimental results and objectively and concisely described what was needed. For this reason, I have no suggestions for improvement.
Based on the above, I believe the article can be accepted for publication after revisions.
In addition to the comments in the sections, below are some recommendations on specific points in the text. It should be noted that the recommendations below are complementary to the previous comments in the sections. Therefore, ALL should be considered as strong recommendations for improvement.
Specific comments:
- Reviewer (Lines 11-26): “Simple Summary” very long and outside the magazine’s standards, because it was written in a very technical and complex manner, making it difficult for non-specialists to understand. The way it was written, it looks more like an "Abstract" than a "Simple Summary", which is not appropriate. Therefore, it should be completely rewritten and adapted to follow the Animals guidelines.
- Reviewer (Lines 11-14): The presentation of the study objective must be appropriate to the scientific method. The one who evaluates, evaluates "if". Ex.: To evaluate "IF" two zilpaterol hydrochloride (ZH) beta-agonists (e.g. Zilmax® (ZM) and Grofactor® (GF)) and a non-supplemented control group (CT), in combination with different meat ageing periods (7, 14, 28, 56 and 120 days) influence the meat quality of South African feedlot bulls (Bos taurus crosses). In addition, the objective in the "Simple Summary" should not be written in a technical way (This should be done in the Abstract). Please provide all necessary improvements.
- Reviewer (Line 61): Words that already appear in the manuscript title (e.g., feedlot; extended ageing; and meat quality) should not be used again in the "keywords". This reduces the chances of the article being found. I strongly recommend that the necessary changes be made, and that the repeated terms be replaced with more specific terms.
- Reviewer (Line 95): What is the meaning of the acronym "NSPCA"? All acronyms should have their meaning defined in the text at the first mention, which was not done. I strongly recommend that you do this.
- Reviewer (Lines 123-125): How much area is available per animal? This must be indicated in the manuscript.
- Reviewer (Lines 125-128): How were the animals weighed? What type of equipment was used? What are the specific characteristics of the weighing equipment? This information should be provided in detail in the manuscript.
- Reviewer (Lines 129-130): Is there a specific reason for this 3-day safety interval before slaughter? Was it carried out to comply with local legislation? It is interesting that this is specified.
- Reviewer (Lines 139-141): Could the results obtained with the CT and ZM treatments have been affected by the bull jumping the dividing fence? If so, to what extent? This should be clearly specified in the manuscript.
- Reviewer (Lines 142-143): It is cited that the bulk feed troughs ensured adequate feeding space per animal. However, the specific space per animal is not specified. This should be clarified in the text.
- Reviewer (Lines 154-155): Again: Was this 3-day period intended to comply with any legislation or standard methodology? If so, which one? This should be specified in the manuscript text.
- Reviewer (Line 156): Is "Slaughter, meat sample collection and storage" a subsection? If so, it should be indicated as 2.1. Slaughter, meat sample collection and storage, in accordance with the Animals author guidelines.
- Reviewer (Lines 167-168): Why were the samples taken from the Longissimus dorsi muscle (LD)? Was there a specific rationale for this? This should be made clear in the text.
- Reviewer (Line 173): Is "Meat quality analyses" a subsection? If so, it should be indicated as 2.2. Meat quality analyses, in accordance with the Animals author guidelines.
- Reviewer (Lines 176-177): Please provide the remaining specifications of the Hunter Lab ColorFlex EZ, such as accuracy, measurement range, response time, etc.
- Reviewer (Lines 185-187): Was this procedure performed using a standard methodology? This should be clearly specified in the text, with reference to the specific methodology used.
- Reviewer (Lines 195-197): The specifications of the Warner-Bratzler instrument (e.g., accuracy, measurement range, response time, etc.) should be provided in the manuscript text.
- Reviewer (Lines 197-198): Why was this done? Wouldn't the sensory panel score be more reliable? The rationale for this choice should be specified in the manuscript text.
- Reviewer (Line 202): Is "Statistical analyses of data" a subsection? If so, it should be indicated as 2.3. Statistical analyses of data, in accordance with the Animals author guidelines.
- Reviewer (Lines 203-212): It would be beneficial to provide a more complete and detailed description of the statistical analyses performed. I recommend doing so.
- Reviewer (Lines 215-222): This passage discusses the results obtained. Therefore, it should be placed in the relevant section, not in the "Results" section.
- Reviewer (Lines 215-222): It is mentioned that some parameters were larger in bulls supplemented with ZH. However, it is not highlighted that there was no statistical difference. Therefore, if the difference is only numerical (and not statistical), this should be explicitly stated in the text to avoid misleading the reader.
- Reviewer (Line 225): The decimal separator must be ".", not ",". Please correct this.
- Reviewer (Lines 230-239): Again: This passage discusses the results obtained. Therefore, it should be placed in the relevant section, not in the "Results" section.
- Reviewer (Line 241): Tables should be self-explanatory. Therefore, all acronyms used should be defined in the legend or supplementary text (below). In addition, the decimal separator should be "." and not ",". Please provide any necessary improvements.
- Reviewer (Lines 254-259 and 260-269): Again: This passage discusses the results obtained. Therefore, it should be placed in the relevant section, not in the "Results" section.
- Reviewer (Lines 277-282): The discussion should always be linked to the results. Therefore, whenever data are discussed, they should be cited to their location in the results (e.g., Table 1, Table 2, or Table 3). I recommend that you check and correct this throughout the text.
----
Best regards,
The reviewer
Author Response
Dear Reviewer,
Thank you for the constructive comments and editorial recommendations. These recommendations were very helpful, and we feel that it has improved the quality and potential impact of the manuscript. Specific comments are addressed briefly below:
Introduction: This section was completely rewritten.
Materials and Methods: More details were added, and specific aspects were better explained as requested.
Results: The results section has been significantly edited according to suggestions. The tables were edited to improve clarity and explanations of what are presented.
Discussion: The 'Discussion' section was significantly edited, and additional figures were added to discuss the results. Additional references were included.
Conclusions: The 'Conclusions' section was edited
Thank you for the specific comments - all of these were addressed and corrected as suggested.
Specific comments:
- “Simple Summary” was shortened and rewritten.
- Key words were edited.
- the acronym "NSPCA" was explained.
- The pens space and feeding space per animal were explained.
- It was explained how animals were weighed - equipment was described. -
- The 3-day safety interval (ZH withdrawal period) before slaughter was explained better.
- The effects of the bull jumping the fence was negligible, as this happened at the start of the study and statistically the use of the Bonferroni method to compensate for unbalanced data, was used to test differences between ZH and CT treatment means.
- The feeding space in the bulk feed troughs was explained.
- Materials and methods were rewritten and sub-headings included to indicate different parts, as recommended.
- An explanation was included that meat samples taken from the Longissimus dorsi muscle (LD), because this is the standard reference muscle generally used in meat science research.
- The specifications of the Hunter Lab ColorFlex EZ, were included.
- It ws explained that standard methodology was used for these meat science analyses and a reference was included.
- Sensory panel analysis was not possible in this study due to the experimental design and limitations in funding. For this reason, the previously published relation between WQBSF and tenderness was used.
- The description of the statistical analyses was edited and more details included.
- The 'Results' and 'Discussion' sections were significantly edited and several paragraphs moved to the discussion section.
- Differences between treatments for several parameters were explained better and that statistical significance indicated in each case.
- The decimal separator was corrected in tables.
- Tables were edited to be self-explanatory.
Reviewer 3 Report
Comments and Suggestions for Authors
Title: Effects of different zilpaterol hydrochloride feed supplements and extended ageing periods on the meat quality of feedlot bulls
The manuscript “Effects of different zilpaterol hydrochloride feed supplements and extended ageing periods on the meat quality of feedlot bulls” evaluated the effects of two commercial zilpaterol hydrochloride (ZH) beta-agonists denoted as β-adrenergic agonists A and B (ZH-A and ZH-B), and a non-supplemented control group (CT), in combination with different meat ageing periods (7, 14, 28, 56 and 120 days) on meat quality of South African feedlot bulls (Bos taurus crosses). It was demonstrated that bulls fed the β-adrenergic agonists ZH-A or ZH-B containing zilpaterol hydrochloride (ZH) for 30 days, significantly decreased meat tenderness through increased WBSF values. It is well written article with some interesting findings; however, there are some corrections before its acceptance for publication:
Line 11-26: The simple summary is very complicated, and how a lay audience can understand such technical language. Authors should avoid describe results here. According to the journal guidelines it should contain a clear statement of the problem addressed, the aims and objectives, pertinent results, conclusions from the study and how they will be valuable to society. This should be written for a lay audience, i.e., no technical terms without explanations. And it must consist of no more than 200 words.
Line 27-60: The abstract portion is too long and it is hard for the readers to extract the useful information. It should give a pertinent overview of the work including background, methods, results and conclusions of the study. Authors must conclude their study at the end of the abstract. Authors must adhere with the journal guidelines for writing the manuscript and abstract must be a single paragraph of about 200 words maximum.
Line 64-99: The introduction part is not clear and need to rewrite. Authors should describe the effect of using growth promoters on meat quality characteristics by citing the previous studies. Also please highlight controversial and diverging hypotheses. Keep the introduction comprehensible to scientists outside your particular field of research.
Line 101-157: These paragraphs should be written under subheading for better understanding of the readers.
Line 164: Which slaughtering method was used? Provide more details.
Line 175: All meat quality parameters such as color, tenderness and water loss must be under separate subheadings.
I was doubt that author just copy and paste the material from the thesis and don’t even bother to format it according to the guidelines of the journal.
The materials and methods part are also lacking the proper format and needs to follow the guidelines of the journal while writing it.
Overall, discussion part is very weak, authors should focus more on what would be possible reasons by which such supplementation influences meat color, tenderness and water losses by giving the relevant references. It was felt that, authors didn’t read properly the previous studies on the topic. Therefore, I invite the authors to read and cite the following previous studies and/or more in order to make the discussion part more interesting:
· https://doi.org/10.3390/ani11092688
· https://doi.org/10.1016/j.livsci.2019.07.017
· http://hdl.handle.net/2263/43564
· Effect of Feeding Zilpaterol Hydrochloride for 20 Days to Calf-fed Holstein Steers With a 3 or 10 Day Withdrawal Period Antemortem on Carcass Characteristics and Tenderness
· https://doi.org/10.2527/jas.2015-9878
· https://doi.org/10.2527/jas.2006-173
· https://doi.org/10.15232/aas.2021-02190
Line 343: Authors should mention what are the benefits of the current study to the meat processors or meat producers? Like they can suggest some guidelines.
Comments on the Quality of English LanguageMinor editing of English language required.
Author Response
Dear reviewer,
Thank you for the constructive criticism and recommendations. We have addressed all of these aspects and think this this will improve the quality and impact of the manuscript.
The manuscript was significantly edited and several sections were rewritten. These indlude rewriting the 'Simple summary' and 'Abstract', and significant editing of the 'Introduction'. The 'Materioals and methods' have been edited and improved. The 'Results' section was edited and the 'Discussion' section mostly rewritten. Additional statistaical data (figures) were included to discuss the findings.
Thank you for the suggestions - these were very valuable.
Sincerely,
Dr. Edward Webb
obo authors
Round 2
Reviewer 2 Report
Comments and Suggestions for Authors
Dear authors,
Thank you for your kind responses and the implemented improvements.
As mentioned earlier, this is a relevant topic, and the conducted study is interesting. While acknowledging that various improvements have been made, some recommendations were not clear in the text or were not followed, such as presenting the study's objective following the scientific method and including specifications of the equipment used, among other aspects. Therefore, I recommend that you carefully review the recommendations provided in the first round of revision and apply the still necessary improvements. My assessment is that the manuscript still requires minor revisions to become suitable for publication.
----
Best Regards,
Reviewer
Author Response
Dear Reviewer 2,
Thank you for the editorial comments and constructive comments. We have edited the manuscript according to your comments and made sure to include the details of the equipment used or to include a relevant reference to describe the methodology better.
A better description of the 'study objective' has been included in Line 81.
In particular we have provided descriptions for the following methods / equipment:
- The 'stunning procedure' is better described in Line 149.
- The El;ectrical stimulation procedure is described in Line 150.
- Details of method, equipment, and accuracy of colour measurements are provided in Line 166.
- Methodology to measure moisture (drip & cooking losses) is presented in Line 176.
- WBSF measurements and apparatus used are presented in Line 184 to 185.
- We trust that these additions are adequate to describe the methods.
Reviewer 3 Report
Comments and Suggestions for Authors
The manuscript is sufficiently improved and may be accepted in present form for its possible publication in Animals.
Author Response
Dear Reviewer 3,
Thank you for the editorial comments and constructive corrections recommended. We have edited the manuscript according to your comments and made sure to include the details of the study objective, equipment used, and included a relevant references to describe the methodology better.
The description of the statistical methods have been improved and more figures have been included in the discussion section to explain the findings in relation to previous studies and with recommendations about industry practices.
Again, thank you for your inputs to make this manuscript better,
Sincerely,
Edward Webb